# Influence of Morphostructural Elements for Buckwheat (*Fagopyrum esculentum* Moench) Productivity in Different Agricultural Systems

**DOI:** 10.3390/plants11182382

**Published:** 2022-09-13

**Authors:** Danuta Romanovskaja, Almantas Razukas, Rita Asakaviciute

**Affiliations:** Lithuanian Research Centre for Agriculture and Forestry, Žalioji a. 2, Trakų Vokė, LT-02232 Vilnius, Lithuania

**Keywords:** buckwheat, morphostructural elements, productivity

## Abstract

The research was carried out at Vokė Branch of the Institute of Agriculture of Lithuanian Research Centre for Agriculture and Forestry in 2018–2019. The objective of this study was to determine the influence of the relative proportion of stems, leaves and flowers on biomass formation and grain yield in organic and conventional farming systems. The study found that buckwheat produced one-third more biomass in the conventional farming system than in the organic farming system. Differences between buckwheat cultivars were more distinct due to biomass formation than grain yield. The research determined that the productivity (biomass and grain yields) of buckwheat was dependent on the proportion of morphological elements in plants and the process of biomass formation and grain yield in organic and conventional agricultural systems. Biomass yields depended on the relative number of stems in both farming systems. Grain yield depended on the ratio of flowers in the morphostructure; however, statistically significant correlations were found only in the organic farming system.

## 1. Introduction

Buckwheat (*Fagopyrum esculentum* Moench) was grown in Asia several thousand years ago and in Europe since the 13th century [1,2]. Recently, the world leaders in terms of buckwheat area are China (37.6% of the total), Russia (22.4%) and Ukraine (9.0%) [3]. Buckwheat is an important agricultural crop because its grains are used for human consumption and in the diet of domestic animals [4]. At the beginning of the 21st century, buckwheat production decreased in East Asian countries and increased in Western European countries [3]. Over the past two to three decades, buckwheat cultivation has declined worldwide owing to the low yields of these crops [5,6]. At times, the lower grain yield may have been due to the technological level of buckwheat cultivation [7]. In addition to the negative aspect of low yields, buckwheat is most suitable for growing on organic farms, as it grows well in a variety of soils without the use of fertilizers or plant protection products. Buckwheat grains are a health-promoting food that is receiving more attention in the functional food production sector because of their high quality proteins, unsaturated fatty acids, minerals, vitamins, antioxidants and phenolic compounds [2,6]. It should be noted that buckwheat grains, unlike other agricultural cereals, are gluten-free, which is very important for gluten-intolerant people [8]. Against this background, buckwheat has good prospects for expanding its crop areas.

Studies on the effects of different agroecological conditions on buckwheat have shown that buckwheat can be successfully grown in different soil types using natural soil fertility [9]. Buckwheat is mostly grown on organic farms, where farming is based on a strong connection with nature and the maintenance of a natural balance [4]. In Central Europe, the average grain yield of organic buckwheat grown in organic crops is 1.24 t ha^−1^, with little variation (CV = 5.05%) [4]. However, according to research by Dutch scientists, high yields of agricultural crops cannot always be expected on organic farms [10]. The average yield difference between conventional and organic farming systems is said to be more than 20%. The choice of varieties for cultivation, even in a conventional farming system where mineral fertilizers are used, is also a very important factor in increasing buckwheat productivity. A study of the influence of mineral fertilizers on the yield of buckwheat cultivars of different morphotypes showed that shorter stem cultivars responded to mineral fertilizers with 15% higher productivity increase [11].

Buckwheat has specific biological properties, as its growth stages overlap throughout the growing season. The flowering stage of buckwheat lasts for approximately two-thirds of the vegetation period, but even with a large number of flowers, the percentage of grain formation may be low [12]. Numerous studies have substantiated that buckwheat productivity is related to bee attendance and hydrothermal conditions during the growing season [12,13,14,15]. The morphostructure of the plant and the formed vegetative biomass become important in buckwheat grain yield formation. Italian researchers state that the number of vegetative structures (stems, branches and leaves) in buckwheat helps to assess the potential yield, as grain yields are directly dependent on biomass [16].

Our research will help to identify changes in the formation of organic and conventionally grown buckwheat yield (biomass and grain) and new patterns. It is likely that buckwheat biomass yield and grain yield depend on different morphostructural elements.

The objective of this study was to determine the influence of the relative proportion of stems, leaves and flowers on biomass formation and grain yield in organic and conventional farming systems.

## 2. Results

The dry matter yield of buckwheat biomass in the organic farming system was on average 26.8% lower than in the conventional farming system (Table 1). However, the grain yields in the two farming systems differed significantly and not every year. According to previous research, the yield index was substantially higher only in organic farming systems.

The results of the research showed that the dry matter yield of buckwheat biomass in the organic farming system was significantly lower (2.75–3.92 t ha^−1^) than in the conventional farming system (3.58–4.99 t ha^−1^), which was on average 27.7% (Figure 1). Only the biomass dry matter yields of cultivars ‘Kvietka’, ‘Canita and ‘Belaruskij’ determinant-2 did not differ significantly between the two farming systems.

Differences in biomass dry matter yield were observed between cultivars grown in different farming systems. In the organic farming system, the biomass dry matter yield of three cultivars (‘Anita Belaruskaya’, ‘Kvietka’, ‘Canita’) was substantially higher than ‘VB Vokiai’ (standard cultivar). In the conventional farming system, as many as eight cultivars had substantially higher biomass dry matter yields.

Grain yield of most studied cultivars (except ‘VB Vokiai’, ‘VB Nojai’, ‘Canita’ and ‘Zaleika’) was higher in the conventional farming system by 3.0–25.2% (Figure 2). Grain yield in the organic farming system was higher in the ‘Kvietka’ cultivar (3.51 t ha^−1^), but there were no significant differences compared to the standard cultivar ‘VB Vokiai’. However, the grain yield of most cultivars was substantially lower compared to ‘VB Vokiai’. In the conventional farming system, the differences in grain yield between cultivars were small, with the exception that cultivars ‘Smuglianka’ and ‘Kvietka’ were substantially higher and ‘Mara’ and ‘Zniajarka’ were generally lower. The grain yield was found to be strongly correlated with the biomass dry matter yield (r = 0.64 ** − 0.76 **) in the organic farming system, independent of the year (Table 2). It should be noted that the correlations found for the conventional farming system were low and statistically unreliable.

The ratio of buckwheat morphological parts (stems, leaves and inflorescences) differed among different farming systems. The relative number of stems was on average 52.3% in the organic farming system and 53.0% in the conventional farming system, i.e., the difference was not significant (Table 3). The relative amounts of leaves and flowers differed more. Compared to the organic farming system, in the conventional farming system buckwheat formed on average 11% more leaves and on average 21% fewer flowers.

The results showed that the yield of buckwheat grain was positively correlated with the relative quantity of stems and flowers, but negatively correlated with the relative quantity of leaves (Table 4). Stronger and more statistically reliable correlations were found in 2018.

The correlations of the yield index with the relative amounts of morphostructural elements exhibited different tendencies. According to the research data, the yield index decreased with an increase in the relative number of stems and leaves. In contrast, as the relative number of flowers increased, the yield index increased. Statistically significant correlations were observed in 2019.

The relative amounts of buckwheat morphostructural elements had different effects on biomass dry matter yield, but correlations showed similar trends in both farming systems estimating two-year average data (Figure 3). Higher stem content (%) in the morphostructure had a greater influence on the increase in biomass dry matter yield. Owing to the increase in the quantity of other morphostructural elements, a decreasing trend in biomass dry matter yield was observed.

Different correlations were found between the morphostructural elements and grain yield. The same morphostructural elements had the opposite effect on buckwheat grain yield depending on the farming system. In the organic farming system, the relative number of stems (if not exceeding 54%) had a greater influence on the grain yield than in the conventional farming system (Figure 4). In the conventional farming system, an upward trend in grain yield was observed due to an increase in the relative number of leaves (if not exceeding 32%). It should be noted that the proportion of flowers in the morphostructure up to 18% also had a positive effect on the grain yield. However, an even higher number of flowers did not result in higher grain yield.

## 3. Discussion

One of the main reason why buckwheat is grown in a small amount compared to other cereals is the low yield, which in Europe averages 0.9 t ha^−1^ [5]. Buckwheat is sensitive to environmental conditions, which results in large variations in grain yield. According to the data from 10-year research conducted in Lithuania, the variation in buckwheat grain yield in the conventional farming system was CV = 24.3–45.5% [15]. This shows that fertilization with mineral fertilizers (N_30_P_50_K_40_) did not result in similar grain yields every year. However, in other experiments in the same farming system, a positive effect on buckwheat biomass yield was observed, increasing by an average of 45.6% due to mineral fertilizer application [16]. The results of our research showed that the dry matter yield of buckwheat biomass in the conventional farming system was 20–33.6% higher than in the organic farming system.

However, not all differences in the biomass dry matter yield of the buckwheat cultivars studied followed these trends. The dry matter yields of ‘Kvietka’, ‘Canita’ and ‘Belaruskij determinant-2′ biomass were similar in both farming systems. It should be noted that the ratio of morphostructural elements of these cultivars also varied slightly, depending on the farming system.

In the conventional farming system, buckwheat has higher biomass dry matter yields owing to the positive effects of mineral fertilization. However, no positive effect of grain yield increase was observed. The grain yield of most buckwheat cultivars grown in the conventional farming system increased on average by 0.27 t ha^−1^. However, the grain yields of the buckwheat cultivars ‘VB Vokiai’, ‘VB Nojai and ‘Canita’ were higher in the organic farming system.

The results of the statistical evaluation showed that the correlation between grain yield and biomass yield was not analogous in either farming system. In the ecological farming system, the amount of biomass had a greater influence (40.1–57.8%) in the process of grain yield formation.

Japanese scientists have found that mineral fertilizers make it easier for plants to grow under favorable climatic conditions. The results of this study showed that grain yields decreased due to excessively high temperatures and precipitation [17].

In Lithuania, buckwheat requires an optimal irrigation hydrothermal regime (HTC 1.0…1.5) at the beginning of vegetation (June) and during the grain maturation period (August), as well as an excessive irrigation hydrothermal regime (HTC 1.5…2.5) during the grain formation period (July) [18]. The results of our research suggest that buckwheat productivity is affected by the hydrothermal regime of each year. In the second year of the study, precipitation was particularly low at the beginning of the vegetation in June (HTC 0.36). Under these conditions, buckwheat plants grew lower and formed less biomass, which had a negative effect on grain yield. Italian researchers have found that buckwheat grain yield is positively correlated with vegetative biomass (r = 0.66 **) [19]. The results of our research confirmed that, in the ecological farming system, the role of biomass was decisive in the process of grain yield formation (R^2^ = 0.41 **–0.58 **).

Buckwheat grown in the conventional agricultural system had a much higher yield of biomass dry matter than that grown in the organic system. Variations in the biomass dry matter and grain yield of the studied varieties were not proportional. This indicates that the yield of buckwheat grains does not always depend on the yield of plant biomass and is influenced by the agricultural systems used.

In the conventional farming system, where plants are better supplied with nutrients, buckwheat grows faster and has more branches, resulting in higher biomass yields. The growth stages of buckwheat occur by overlapping the entire vegetation, i.e., plants grow and form vegetative and generative organs simultaneously. An excessively humid hydrothermal regime, which was favorable for vegetative growth, was unfavorable for flower pollination and grain formation. The results of our research showed that in 2018, when the hydrothermal regime in August was irrigation excessive (HTK 2.12), the grain yield was lower, on average by 17.3–29.9%. Accordingly, the yield index was lower.

The morphostructural element content (%) in buckwheat plants had different effects on buckwheat productivity (biomass and grain yield). A strong correlation was found between stem content and biomass dry matter yield: in the organic farming system, R^2^ = 0.67 ** and in the conventional farming system, R^2^ = 0.41 **. Other morph-structural elements, such as leaves and flowers, have an influence on biomass dry matter partiality decline. The increase in the relative amounts of other morphostructural elements (leaves and flowers) influenced the decreasing trends in biomass dry matter.

This can be explained by the fact that the relative number of stems accounted for a larger share in the morphostructure of buckwheat plants than that in the leaves or flowers.

Grain yield correlated more strongly with the relative number of flowers. Trends in grain yield increased with an increasing relative number of flowers in the plants in each study year. However, the roles of all morphostructural elements in buckwheat grain yield formation were the same in both the study years. One could determine the relative number of flowers on which the buckwheat grain yield depended, especially in the organic farming system. The dependence of buckwheat grain yield on the relative amount of flower formation did not change considerably in each study year.

The correlations between individual morphostructural elements and yield index showed different trends because of their influence on the size of the yield index. An increase in the relative number of stems and leaves led to a decrease in the yield index. In contrast, an increase in the relative number of flowers led to an increase in the yield index. Stronger correlations between morphostructural elements and yield index were found in the study years, where precipitation was 50% lower than the standard climatic norm. The results of this study showed that, when there was a lack of precipitation during the growing season, the yield index depended more on the relative quantity of morphostructural elements.

The yield index describes the ratio of grain yields to total plant biomass and it formed less in summer when less precipitation occurred. We believe that because of the formation of less biomass due to unfavorable abiotic factors, the formation of buckwheat grain yield was determined by a better distribution of photosynthetic products between vegetative structural elements (stems and leaves) and reproductive structural elements (inflorescences and grains).

The pattern of buckwheat crop formation may vary depending on the cultivar. The main factor in the increase in buckwheat yield is the better distribution of assimilates in determinant-type buckwheat cultivars with limited branching and fewer leaves [20]. According to the results of our study, the role of leaves formed in buckwheat plants in the process of grain yield formation is less important.

## 4. Materials and Methods

### 4.1. Site and Soil

A field experiment was performed in the crop rotation of the Voke Branch of Institute of Agriculture of the Lithuanian Research Centre for Agriculture and Forestry, which is located in Trakų Voke (54°33′ N, 25°10′ E), during 2018–2019. The experimental plots were established on sandy loam on carbonaceous fluvial-glacial gravel-eluviated soil (IDp), according to the FAO-UNESCO classification of Haplic Luvisols (LVh) [17]. Soil agrochemical characteristics: pH_KCl_—5.2–6.2, mobile P_2_O_5_, 108–152 mg kg^−1^; mobile K_2_O, 150–165 mg kg^−1^ and humus 2.11–2.18%.

### 4.2. Experimental Design and Management

Fifteen buckwheat (*Fagopyrum esculentum* Moench) 15 cultivars were studied in a two-factor experiment: Factor A—farming systems: (1) organic and (2) conventional. Factor B—buckwheat cultivars: (1) ‘VB Vokiai’ (LT), (2) ‘VB Nojai’ (LT), (3) ‘Kora’ (PL), (4) ‘Panda’ (PL), (5) ‘Volma’ (BY), (6) ‘Smuglianka’ (BY), (7) ‘Anita Belaruskaya’ (BY), (8) ‘Anika’ (BY), (9) ‘Kvietka’ (BY), (10) ‘Canita’ (BY), (11) ‘Zaleika’ (BY), (12) ‘Mara’ (BY), (13) ‘Zniajarka’ (BY), (14) ‘Belaruskij determinant-1’ (BY), (15) ‘Belaruskij determinant-2’ (BY). These cultivars were moderately late. In order to preserve organic matter in the soil, especially in the organic farming system, crop rotation was applied. In both research years, buckwheat was sown after bean plants—lupins (*Lupinus angustifolius* L.). The soil for buckwheat trials was cultivated twice in spring. Fertilization and other measures: zero application in the organic farming system and N_40_P_60_K_60_ (N 34.4% (Achema, Lithuania), P_2_O_5_ 19% (Emolus, Poland), K_2_O 60% (Emolus, Poland) and used plant protection products in conventional farming. The two-factor experiment was performed with four replicates. The test plot area for cultivar trials was 2 m^2^. Seed rate: 3 million ha^−1^ of fertile seeds. Buckwheat was sown during the 3rd ten-day period of May each year. Under the climatic conditions of Lithuania, buckwheat bloomed intensively and formed mature grains in July and August. Harvesting took place during the 1st ten-day period of September.

### 4.3. Meteorological Conditions

Meteorological conditions were assessed based on data provided in the bulletins of the Lithuanian Hydrometeorological Service by calculating the hydrothermal coefficient (HTC). The HTC is the ratio between the sum of precipitation over a period of time and the sum of the daily mean temperature (≥10 °C). According to the long-term average meteorological conditions, the average temperature of the summer season in the Traku Voke area is 16.9 °C, precipitation is 237 mm and, according to the hydrothermal coefficient, the season irrigation is exceedingly high (HTC > 1.50). July is 0.9–2.3 °C warmer and 16–18% wetter in summer than June and August. The hydrothermal regime favorable for the growth of agricultural plants is optimal irrigation (HTC 1.0 … 1.5).

The meteorological conditions during the summer season in the year of the experiment were different from the long-term average. In both years, there was a lack of precipitation in June 2019, with a very heavy drought hydrothermal regime in June (HTC 0.36) (Table 5). The hydrothermal regimen in August differed from the optimal regimen in both the study years. August 2018 was wet (HTC 2.12) and August 2019 suffered a heavy drought (HTC 0.42). The July hydrothermal regime in 2018 and 2019 was sufficiently moist and optimal for buckwheat growth.

### 4.4. Biomass and Grain Harvesting

The buckwheat biomass yield and morphological elements’ (stems, leaves and flowers) quantity was determined at the flowering stage (BBCH 60) [21], until there were no seeds, that is, in July. Plants determined for buckwheat biomass were cut in each field from an area of 1 m^2^, leaving a remaining area of 1 m^2^ to determine the buckwheat grain yield. The ratio of morphological parts was determined by taking 10 plants from each field. Leaves and flowers were separated from the stems, dried and dry matter content was determined. Their relative amount was determined by weight method. Grain harvesting was performed in September.

### 4.5. Statistical Analyses

Two-way analysis of variance with the use Fisher’s test and significance of differences (LSD_05_) were determined at the statistical significance level of *p* ≤ 0.05. Correlation analysis was performed using of Excel program. Linear (on the table) and nonlinear (on the figure) correlation coefficients were calculated between variables (*n* = 15) and marked as non-significant (ns), with the significance of coefficients being *p* ≤ 0.05 * and *p* ≤ 0.01 **.

## 5. Conclusions

In the conventional farming system, buckwheat had a significantly higher biomass dry matter yield (*p* < 0.05). Biomass dry matter yields varied more among cultivars and were higher in conventional farming than in organic farming. Buckwheat grain yields in different farming systems did not differ significantly among the different cultivars.

In the ecological farming system, buckwheat forms relatively more generative morphostructural elements (flowers) but fewer vegetative elements (stems and leaves) than in the conventional farming system.

The dry matter yield of buckwheat biomass was significantly (*p* < 0.01) dependent on the relative quantity of stems in both farming systems.

Buckwheat grain yield was significantly (*p* < 0.01) dependent on the relative number of flowers in the organic farming system. In the conventional farming system, trends in buckwheat grain yield with an increasing relative number of flowers have been observed.

Higher yield indices and reliable (*p* < 0.05, *p* < 0.01) correlations with morphostructural elements were present in the presence of less precipitation during the growing season.

## Figures and Tables

**Figure 1 plants-11-02382-f001:**
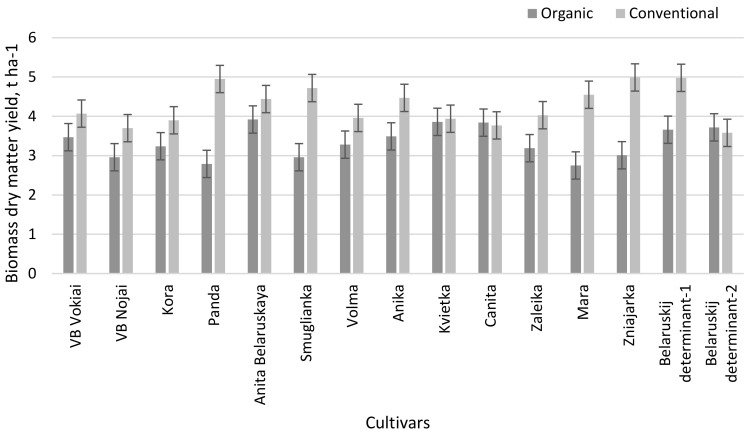
Biomass yield of buckwheat cultivars in different farming systems (LSD_05(AxB)_ = 0.347).

**Figure 2 plants-11-02382-f002:**
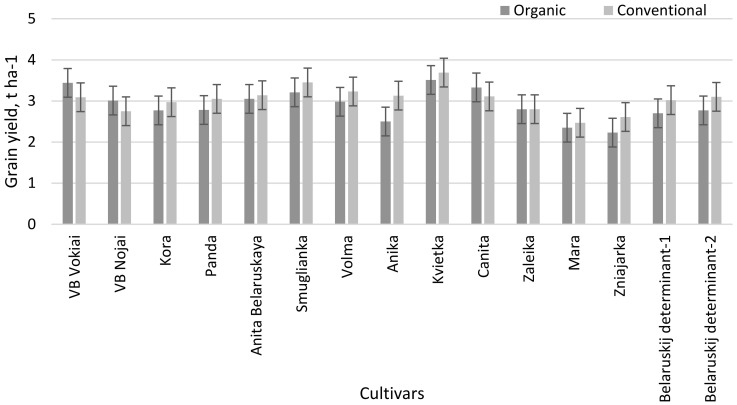
Grain yield of buckwheat cultivars in different farming systems (LSD_05(AxB)_ = 0.349).

**Figure 3 plants-11-02382-f003:**
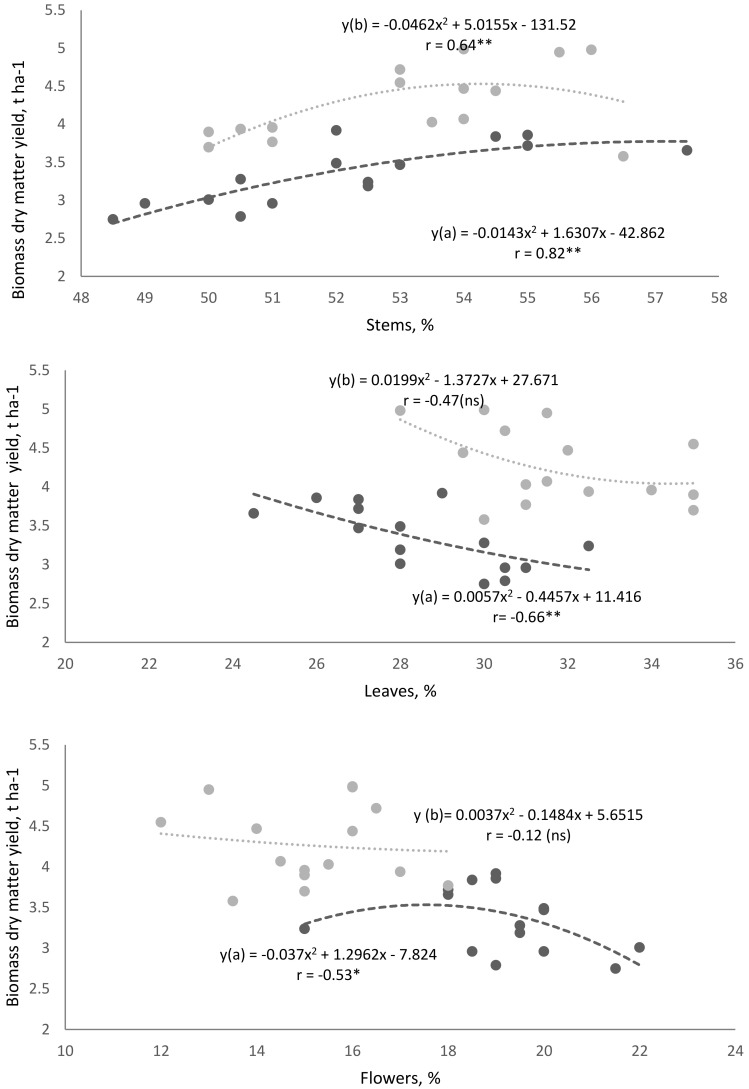
Correlation between morphostructural elements of buckwheat plants and biomass dry matter yield: a—organic farming system, b—conventional farming system (*—Significant values with *p* < 0.05; **—Significant values with *p* < 0.01; ns — not significant).

**Figure 4 plants-11-02382-f004:**
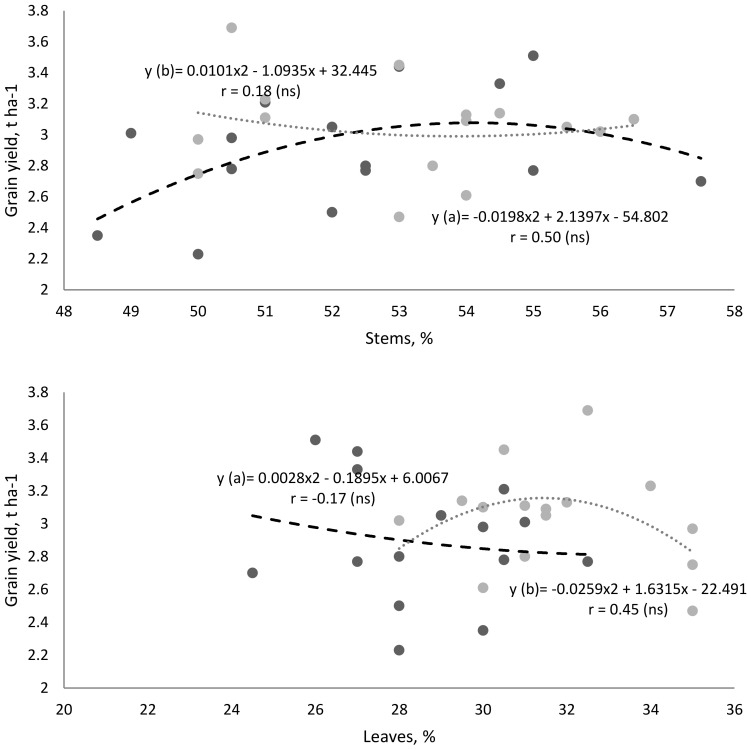
Correlation between morphostructural elements of buckwheat plants and grain yield: a—organic farming system, b—conventional farming system (**—Significant values with *p* < 0.01; ns — not significant).

**Table 1 plants-11-02382-t001:** Yields and harvest indexes of biomass and grain for organic and conventional farming systems.

Indicator	Year	Indicator Values in Different Agricultural Systems		LSD_05(A)_
		Organic	Conventional	
Biomass dry matter, t ha^−1^	2018	3.78	5.05	0.100
	2019	2.90	3.48	0.081
Grain yield, t ha^−1^	2018	2.66	2.64	0.079
	2019	3.12	3.43	0.102
Harvest index	2018	0.41	0.35	0.009
	2019	0.52	0.49	0.011

**Table 2 plants-11-02382-t002:** Correlation between yield of buckwheat biomass dry matter (t ha^−1^) and grain yield (t ha^−1^) (*n* = 15).

Year	Correlation Coefficient (r)	
	Organic Farming System	Conventional Farming System
2018	0.64 **	0.47 (ns)
2019	0.76 **	0.19 (ns)

**—Significant values with *p* < 0.01; ns—not significant.

**Table 3 plants-11-02382-t003:** Relative content (%) of morphostructural elements in plants of different buckwheat cultivars in organic and conventional farming systems.

Cultivars	Organic			Conventional		
	Steams	Leaves	Flowers	Steams	Leaves	Flowers
‘VB Vokiai’	53.0 ± 6 *	27.0 ± 1	20.0 ± 6	54.0 ± 4	31.5 ± 1	14.5 ± 4
‘VB Nojai’	49.0 ± 5	31.0 ± 1	20.0 ± 5	50.0 ± 3	35.0 ± 2	15.0 ± 4
‘Kora’	52.5 ± 7	32.5 ± 3	15.0 ± 4	50.0 ± 9	35.0 ± 2	15.0 ± 7
‘Panda’	50.5 ± 9	30.5 ± 3	19.0 ± 6	55.5 ± 4	31.5 ± 1	13.0 ± 4
‘Anita Belaruskaya’	52.0 ± 7	29.0 ± 2	19.0 ± 5	54.5 ± 5	29.5 ± 2	16.0 ± 6
‘Smuglianka’	51.0 ± 5	30.5 ± 1	18.5 ± 5	53.0 ± 8	30.5 ± 2	16.5 ± 6
‘Volma’	50.5 ± 7	30.0 ± 1	19.5 ± 7	51.0 ± 7	34.0 ± 3	15.0 ± 4
‘Anika’	52.0 ± 5	28.0 ± 1	20.0 ± 5	54.0 ± 2	32.0 ± 1	14.0 ± 1
‘Kvietka’	55.0 ± 4	26.0 ± 1	19.0 ± 4	50.5 ± 6	32.5 ± 1	17.0 ± 5
‘Canita’	54.5 ± 6	27.0 ± 1	18.5 ± 6	51.0 ± 8	31.0 ± 3	18.0 ± 5
‘Zaleika’	52.5 ± 8	28.0 ± 2	19.5 ± 7	53.5 ± 5	31.0 ± 1	15.5 ± 4
‘Mara’	48.5 ± 10	30.0 ± 2	21.5 ± 8	53.0 ± 6	35.0 ± 4	12.0 ± 2
‘Zniajarka’	50.0 ± 7	28.0 ± 1	22.0 ± 8	54.0 ± 7	30.0 ± 1	16.0 ± 6
‘Belaruskijdeterminant—1′	57.5 ± 7	24.5 ± 2	18.0 ± 5	56.0 ± 7	28.0 ± 2	16.0 ± 5
‘Belaruskij determinant—2′	55.0 ± 8	27.0 ± 2	18.0 ± 5	56.5 ± 4	30.0 ± 1	13.5 ± 4

*—±SE (standard error).

**Table 4 plants-11-02382-t004:** Correlation coefficients (r) between relative proportions of morphological elements and grain yield and harvest index each year (*n* = 15).

Morphostructural Elements	Coefficients of Correlation (r)			
	Grain Yield		Harvest Index	
	2018 Year	2019 Year	2018 Year	2019 Year
Organic farming system				
Stems, %	0.60 *	0.44 (ns)	−0.43 (ns)	−0.46 (ns)
Leaves, %	−0.72 **	−0.10 (ns)	−0.46 (ns)	−0.65 **
Flowers, %	0.50 *	0.68 **	0.45 (ns)	0.57 *
Conventional farming system				
Stems, %	0.51 *	0.35 (ns)	−0.21 (ns)	−0.61 *
Leaves, %	−0.71 **	−0.40 (ns)	−0.31 (ns)	−0.80 **
Flowers, %	0.38 (ns)	0.33 (ns)	0.34 (ns)	0.60 *

*—Significant values with *p* < 0.05; **—Significant values with *p* < 0.01; ns—Not significant.

**Table 5 plants-11-02382-t005:** Hydrothermal regime in Traku Voke area during summer season.

Month	HTC		
	2018	2019	Long Term Average
June	1.10	0.36	1.55
July	1.09	1.27	1.59
August	2.12	0.42	1.41
Summer season	1.45	0.68	1.52

## Data Availability

Data is contained within the article.

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
