# Peer review of "Influence of Morphostructural Elements for Buckwheat (Fagopyrum esculentum Moench) Productivity in Different Agricultural Systems"

_plants, 2022, doi:10.3390/plants11182382_

Round 1

Reviewer 1 Report

The aim of study was to determine the influence of the relative proportion of stems, leaves, and flowers on buckwheat biomass formation and grain yield in organic and conventional farming systems. The topic fits to requirements of journal PLANTS. The obtained data on buckwheat plant development and influence of morphological parts on biomass and grain yield in two farming systems could explain formation of yield and differences of crop development under different farming systems.  However, used methodology was not described very properly, so data interpretation would be improved, even if simplest methods of data analyses were applied.

It was uncomfortable to reed part of Materials and methods after Results and Discussions, but it is requirement of journal. The first table appearing in article was numbered as Table 2 probably in relation to this, but instructions to authors indicates, that “…Tables should be inserted into the main text close to their first citation and must be numbered following their number of appearance…”.

Introduction describes buckwheat production and explains aim of study. The quality of article will benefit if more recent publications would be used in citations, now the latest are from 2017.

The materials of trials are described sufficiently. Concerning methodology, the explanation – how the ratio of morphological parts was detected, using weight or number of parts, DW or FW – has to be included. The part of statistical analyses has to be specified, the “anova” means “analysis of variance” and for calculation of this different software’s could be used, even excel program. It would be good to specify which coefficient of correlation was used. According to figures in article, some elements of regression analyses were applied, it is recommended to describe those.

 The careful using of concept or designation is recommended. The two factor experiment was described, factor A – farming system: organic and conventional. Further in text appears meaning “ecological”, but in figures this factor is designated as “no fertilisers” and “N40P60K60”, it makes reviewer confused – is the topic about farming systems or fertilisation regime.

The results are described quite widely, just sometimes previously mentioned imprecise usage of designations makes confusions.

Many citations and comparisons with results of other researchers was included in discussions, it is commendable, only more resent publications could be recommended for citation.

More detailed explanation is recommended for relationship between ratio of flowers and grain yield in both farming systems. May be some phenological factors affect grain formation not only climatic conditions. For instance – high N supply promotes and prolongs vegetative part development, delaying development of reproductive parts of plant.

Factor B in the applied design of experiment was buckwheat cultivars, the influence of genotype on detected traits would be detected easily using Anova. This would be great supplement to explanations like - “The pattern of buckwheat crop formation may vary depending on the cultivar” – page 9.

The article would benefit from careful language editing, several carelessness errors appear, as well as stylistic mistakes should be improved, for instance:

“The main reason for the low yield of buckwheat compared to other cereals is the low yield…” – page 8,

“Differences in grain yield were smaller because of different agricultural systems.” – may be between? – page 3,

“The soil for buckwheat trials was plowed in autumn, cultivated twice, and harvested (?) in spring.” – page 10.

Etc.

Author Response

We took into consideration reviewer's remarks.

Reviewer 2 Report

English and used scientific terms must be significantly revised;

I suggest do not use therm grains for buchwheat seeds, because these term is applied to cereals.

Introduction: the introduction did not convince me as a reader of the topics relevance.

Materials and methods -tested traits and methods for anlyzing them are very poorly described.

Approaches used for statistical data analysis must be reviewed.

Results analysis must be improved

Author Response

(The authors gave the same response as above.)

Reviewer 3 Report

The research attempted to compare the growth of buckwheat in organic and conventional soils. Major issues have to do with

A. Text organization --- 1. Introduction, 2. Results , no #3, 4. Materials and Methods, 5. Conclusions. This order of presentation is faulty and must be corrected to follow the norm --- Introduction, Materials and Methods, Results & Discussion or Results, Discussion, and Conclusions

B. What are the real differences between the organic and conventional plots? A soil analytical table is necessary to show the soil fertility and management systems for the two soils. If the study was conducted on the same soil and the only difference was that one received inorganic fertilizers and the other did not, this will simply be a comparison between a fertilized and non-fertilized soil. There must be something more concrete to describe one as organic and the other conventional

C. What are the authors' definition for biomass? Why should grain yield be separated from biomass? The popular definition for biomass includes the entire biological mass (organic tissue) produced by the plant including foliage and grain yields. Authors presented the two as different entities, and this is wrong.

Author Response

(The authors gave the same response as above.)

Round 2

Reviewer 1 Report

Thank you for improved details, the article looks better. and is easier comprehensible. But I strongly recommend to delet words "using software Anova" in page 11. As it was explained before, ANOVA is abreviation of "analysis of variance" and not a software programme. Please be more accurate in future as well.

Author Response

We are honored to submit the revised manuscript entitled “Influence of morphostructural elements for buckwheat (Fagopyrum esculentum Moench) productivity in different agricultural systems” (1881601) for publication Plants. We took into consideration reviewer’s remarks and marked.

We retained the statistical part and deleted the inscription "ANOVA".

We kept the English language as much as we could.

We wish you a nice time reading this new manuscript and hope you will find it suitable for publication in Plants. I remain available for any question if necessary.